# Microbial Pattern in Amniotic Fluid from Women with Premature Rupture of Membranes and Meconium-Stained Fluid

**DOI:** 10.3390/ph18010037

**Published:** 2024-12-31

**Authors:** Fauna Herawati, Patricia Valery Rahaded, Ruddy Hartono, Rika Yulia

**Affiliations:** 1Department of Clinical and Community Pharmacy, Faculty of Pharmacy, University of Surabaya, Surabaya 60293, Indonesia; fauna@staff.ubaya.ac.id (F.H.); usmanyvalery@gmail.com (P.V.R.); 2Center for Medicines Information and Pharmaceutical Care (CMIPC), Faculty of Pharmacy, University of Surabaya, Surabaya 60293, Indonesia; 3Department of Pharmacy, Rumah Sakit Bhayangkara H.S. Samsoeri Mertojoso Surabaya, Jalan Ahmad Yani No. 116, Surabaya 60231, Indonesia; rudd_apt@yahoo.com

**Keywords:** infections, pathogenicity, amniotic fluid, fetal membranes, premature rupture, infant, newborn, risk

## Abstract

Intra-amniotic infection (IAI), also known as chorioamnionitis, is a major cause of maternal and neonatal infection that occurs during pregnancy, labor and delivery, or in the postpartum period. Conditions such as meconium-stained amniotic fluid (MSAF) and premature rupture of membranes (PROMs) are recognized risk factors for amniotic fluid infection. This study identifies the microbial patterns in the amniotic fluid of women with PROMs and MSAF to determine the presence and types of bacterial growth. It also identifies trends in antibiotic use through descriptive statistics. Conducted as a descriptive observational study with prospective data collection, this research included maternal patients with PROMs lasting more than 12 h and those with MSAF, along with their infants. Of 30 cultured amniotic fluid samples, bacterial growth was observed in 13 cases, with *Escherichia coli* being the most prevalent (40%). Infants born with PROMs accompanied by MSAF were 5.5 days, significantly longer than those born with PROMs alone (3.19 days) or MSAF alone (3.91 days), with a significant difference between groups (*p* = 0.003). In addition, *Escherichia coli* isolates in this study are resistant to ceftriaxone, a third-generation cephalosporin antibiotic. Understanding these microbial patterns is critical for guiding clinical decisions, particularly in managing the risk of infection in pregnant women with PROMs and MSAF and ensuring better outcomes for both mothers and newborns.

## 1. Introduction

Infectious diseases are among the leading complications, responsible for approximately 75% of all maternal deaths [1]. In East Java Province, maternal mortality due to infections is expected to increase from 0.38% in 2019 to 1.42% in 2020 [2]. Infections during pregnancy significantly increase the risk of fetal infection, which can occur in utero, during labor, or postnatally [3]. Intra-amniotic infection (IAI), also known as chorioamnionitis, is a prominent cause of such infections. IAI is characterized by acute infection and inflammation of the chorionic and amnionic layers of the fetal membrane, amniotic fluid, and placental deciduas [4]. Amniotic fluid plays a critical role in fetal development, protecting the fetus from mechanical trauma and infection due to its antibacterial properties. It also provides the necessary environment for the growth and development of fetal organ systems, including the musculoskeletal, digestive, and respiratory systems [5]. Risk factors for IAI include the presence of meconium in the amniotic fluid and premature rupture of membranes (PROMs) [6]. Meconium (the fetal fecal matter)-containing amniotic fluid occurs in approximately 1 in 10 births [7,8,9]. It is crucial to recognize that amniotic fluid can contain ammonium when the fetus experiences stress [10]. Postnatal management must focus on promptly clearing meconium from the baby’s airway, as low Apgar scores are directly linked to a significantly higher risk of mortality [1,10,11]. The presence of meconium in the amniotic fluid may facilitate bacterial growth by acting as a growth medium, potentially inhibiting the natural bacteriostatic properties of the fluid or compromising immune defense mechanisms, thereby increasing the risk of IAI [12]. This infection typically occurs due to the ascent of microorganisms from the lower genital tract [4]. Studies show that common vaginal bacteria, such as Ureaplasma urealyticum, *Escherichia coli*, and *Streptococcus agalactiae*, are often found in the amniotic fluid of women with IAI [13]. Pregnant women with meconium-stained amniotic fluid (MSAF) have a higher prevalence of positive amniotic fluid cultures and an increased risk of neonatal sepsis and mortality [14]. Therefore, antibiotic therapy is recommended for both mothers and newborns during delivery in cases of MSAF and premature rupture of membranes (PROMs) [7,15].

Judicious use of antibiotics in newborns is imperative to prevent antibiotic resistance. According to Darwin’s theory of evolution, natural selection favors traits that enhance survival, leading to antibiotic resistance in bacteria [16]. Identifying the specific bacteria causing an infection guides appropriate antibiotic use, preventing overuse of broad-spectrum antibiotics and minimizing resistance risk. Meconium itself is not a risk factor for infection [1,17,18]; antibiotics should only be administered when there is clear evidence of infection [19]. A review reported that narrow or broad-spectrum antibiotics have similar efficacy in endometritis, febrile morbidity, wound infection, and urinary tract infection in mothers after postoperative cesarean sections. However, there are no data available on infant outcomes [20]. Overuse in newborns can cause serious complications like bronchopulmonary dysplasia [21] and dysbiosis [22,23], which disrupt immune function and increase the risk of metabolic disorders. This study highlights the concerning trend of high broad-spectrum antibiotic use in mothers and newborns, emphasizing the need for evidence-based clinical practice to use antibiotics only for confirmed infections.

Currently, in East Java, especially in Surabaya, there is a lack of data regarding microbial patterns and antibiotic use in patients with MSAF and PROM, which pose a potential risk of infection to both mothers and newborns. This study aims to determine the microbial patterns and antimicrobial susceptibilities in the amniotic fluid of mothers delivering with MSAF and PROMs at a public referral hospital in Surabaya and provide data on maternal and neonatal outcomes after delivery. This surveillance study identifies trends in antibiotic use through descriptive statistics. A correlation analysis was not performed as the study was not designed to identify infection risk factors.

## 2. Results

The predominant age group in this study was 26 to 35 years, with the youngest participant being 24 and the oldest 43. The majority were term pregnancies. There were three cases of fetal malpresentation. Preterm premature rupture of membranes (PROMs) was observed in 17 cases, while meconium-stained amniotic fluid (MSAF) occurred in 11 cases. In addition, there were two cases of PROMs with MSAF. Table 1 reports the characteristics of the samples.

Of the 30 amniotic fluid samples analyzed, bacterial growth was identified in 13 samples (43%). Gram-negative bacteria were predominant (Figure 1), accounting for 11 positive cultures (73.33%), with *Escherichia coli* being the most commonly isolated, found in six samples (40%). In addition, extended-spectrum beta-lactamase (ESBL) production in *Escherichia coli* was identified in two samples (13.33%), followed by *Klebsiella pneumoniae* in two samples (13.33%) and *Enterobacter* spp. in one sample (6.67%). Gram-positive bacteria were isolated in four cases, including *Staphylococcus hemolyticus* (13.37%), *Staphylococcus epidermidis* (13.37%), *Streptococcus beta-hemolyticus* (13.37%), and *Enterococcus* spp. (13.37%). Of note, two samples with MSAF showed polymicrobial growth with the presence of *Escherichia coli* and *Klebsiella pneumoniae*. ESBL production in *Escherichia coli* were identified in specimens with Grade II and Grade III MSAF. Laboratory results indicated a significant immune response in both neonates and mothers, particularly in cases involving Gram-negative bacteria such as *Escherichia coli* and *Klebsiella pneumoniae*. Neonates with MSAF, especially those with concurrent PROMs, had significantly elevated CRP levels, suggesting severe inflammation or sepsis. Across all infections, neonates and mothers had elevated leukocyte and neutrophil counts, reflecting a robust inflammatory response. Extended-spectrum beta-lactamase (ESBL) production in *Escherichia coli* in PROMs with MSAF cases further underscores the severity of infection in these scenarios. Data are presented in Table 2 and Table 3. Further details on the distribution of bacterial growth and the prevalence of Gram-negative bacteria can be seen in Figure 2, which illustrates the defined daily dose per 100 bed-days of antibiotics in neonates.

## 3. Discussion

Antibiotics are used against bacteria; therefore, aside from indication, antibiotic choices are associated with the type of bacteria and its sensitivity to antibiotics. A systematic review of 3728 pregnant women diagnosed with premature rupture of membranes (PROMs) in China identified 1706 microbial isolates. The results showed a predominance of Gram-positive bacteria (54%), followed by Gram-negative bacteria (23%). The most commonly isolated bacteria were *Staphylococcus* (*n* = 643), *Escherichia coli* (*n* = 204), *Enterococcus* (*n* = 99), *Lactobacillus* (*n* = 78), *Enterobacter* (*n* = 61), and *Streptococcus* (*n* = 60) [24]. In contrast, a study by R. Romero et al. in Detroit, USA, involving 59 women with singleton pregnancies diagnosed with PROMs found that *Sneathia amnii* (28.5%) and *Ureaplasma* spp. (14.3%) were the most common bacterial isolates from amniotic fluid cultures. A cross-sectional study conducted at the Hospital of Sótero del Río in 108 pregnant women divided into two groups—those with meconium-stained amniotic fluid (MSAF) (*n* = 64) and those with clear amniotic fluid (*n* = 42)—showed that Gram-negative bacteria were the most common microorganisms (*n* = 4), followed by Gram-positive bacteria (*n* = 2) and *Mycoplasma hominis* (*n* = 1) [25]. Similarly, a study by Rini in Semarang, Indonesia, found that *Escherichia coli* was the most commonly isolated bacterium in MSAF cases, with 25.7% in vaginal deliveries and 5.7% in cesarean deliveries [26]. The differences in microbial patterns observed between this study and others conducted in different geographical locations may be attributed to several factors, including genetic differences influencing immune responses, regional variations in normal microflora, diet, environmental factors, and hygiene practices [27,28].

This study examined cases of PROMs lasting more than 12 h. According to a study by Yin H et al. of 102 women with singleton pregnancies and PROMs, the incidence of intra-amniotic infection (IAI) and neonatal sepsis increased with the duration of membrane rupture. The microbial diversity in the amniotic fluid increased significantly at 12 h after rupture, with further increases noted at 24 h. These findings suggest that microbes can invade the placenta within 12 h and reach the amniotic cavity by 24 h, highlighting the need to consider ascending pathways of infection in PROMs cases [29]. Furthermore, a study at Cipto Mangunkusumo Hospital in Jakarta showed that the risk of neonatal sepsis (Odds Ratio) was significantly higher in cases where PROMs lasted ≥18 h before hospital admission (OR 3.08), ≥15 h during hospitalization (OR 7.32), and ≥48 h before delivery (OR 5.77) [30]. In a prospective cohort study of 200 pregnant women with PROMs whose neonates were evaluated for sepsis after birth, the major outcomes of birth asphyxia (8%), neonatal sepsis (4%), NICU admission (26%), and neonatal mortality (2%) were analyzed in relation to time from PROMs. Neonatal sepsis rates increase after 37 h of PROMs latency [31].

Table 4 shows that the incidence of premature rupture of membranes (PROMs) was higher than that of meconium-stained amniotic fluid (MSAF), with 17 cases compared to seven cases. Similarly, a 2017 study at RSUD Ungaran reported that PROMs was one of the most common complications, accounting for 43.1% of labor-related problems [32]. Table 3 shows that when comparing amniotic fluid status with APGAR scores at 1 and 5 min, the mean APGAR scores were below 7 for all groups, with the lowest scores observed in the MSAF group. However, analysis showed no significant differences in APGAR scores between the PROMs, MSAF, and PROMs with MSAF groups (*p* = 0.687). These results are consistent with a study from Karsa Husada Hospital, where no significant differences were found between APGAR scores and membrane conditions (*p* = 0.638) [33]. Conversely, a study by Masood et al. reported that 77.4% of newborns with MSAF had an APGAR score less than 6, with a significant difference compared to those born with clear amniotic fluid (*p* = 0.002) [34]. To provide additional context on antibiotic management in neonates, Table 5 presents the defined daily dose per 100 bed-days of antibiotics.

The mean length of hospital stay for infants born with PROMs and MSAF was 5.5 days, significantly longer than those with PROMs alone (3.19 days) or MSAF alone (3.91 days), with a significant difference between groups (*p* = 0.003). PROMs and MSAF are recognized risk factors for neonatal infection. In the case of MSAF, the prolonged length of stay can be attributed to the need for respiratory support and the risk of meconium aspiration syndrome (MAS) [35,36]. In addition, PROMs and MSAF are risk factors for early-onset neonatal sepsis (EOS), which may also contribute to prolonged hospital stay [37]. Regarding gestational age, no significant differences in APGAR scores or length of hospital stay were observed between different gestational age groups in this study. This result is in contrast to the findings of Tavares et al., who showed a significant difference in APGAR scores between different gestational age groups (*p* = 0.021) [38].

In this study, Gram-negative bacterial isolates showed the highest antibiotic susceptibility to amoxicillin–clavulanic acid, piperacillin–tazobactam, amikacin, chloramphenicol, and meropenem, all with 100% susceptibility (11/11). Ampicillin–sulbactam, ceftazidime, and ciprofloxacin followed with 91% sensitivity (10/11), and cefotaxime, cefepime, ceftriaxone, and gentamicin each had 82% sensitivity (9/11). Ampicillin had a slightly lower sensitivity of 73% (8/11). For Gram-positive isolates, antibiotics with 100% sensitivity included chloramphenicol, erythromycin, levofloxacin, linezolid (all 4/4), doxycycline, azithromycin, ciprofloxacin, clindamycin (all 3/3), ampicillin, cefoxitin, trimethoprim–sulfamethoxazole, gentamicin (all 2/2), and cefotaxime, cefepime, ceftriaxone, and teicoplanin (all 1/1). Zeng et al. reported similar findings in cases of PROMs, where Staphylococcus isolates were resistant to most penicillins except oxacillin, but were susceptible to first- and second-generation cephalosporins and aztreonam [25]. In addition, a study conducted at RSUD Saiful Anwar Malang on mothers with PROMs found that coagulase-negative staphylococci were most sensitive to amoxicillin–clavulanic acid, fosfomycin, and amikacin [37]. *Escherichia coli* in this study also showed high susceptibility to amoxicillin–clavulanic acid, which is consistent with our findings.

Antibiotic surveillance raises the alarm about the inappropriate use of antibiotics to manage the risk of infection in babies born with meconium-stained amniotic fluid [18]. The use of broad-spectrum antibiotics is increasing, pediatricians are prescribing ampicillin–sulbactam instead of ampicillin, and the use of meropenem is increasing from 0.14 to 0.19 DDD per 100 bed-days. This study reports on the incidence of ESBL bacteria in two out of 11 isolates. The Indonesian regulation proposed that physicians consult the antibiotic stewardship team for antibiotics in the Reserve category; however, meropenem is today in the Watch category and ampicillin/sulbactam is in the Access category.

As reported in the Section 2, only 43% of isolates grew, which is a limitation of this study. These microbiological examinations were performed because prescribers preferred broad-spectrum antibiotics to narrow-spectrum antibiotics. These microbiological findings will encourage prescribers to use narrow-spectrum antibiotics.

## 4. Materials and Methods

This study is a descriptive observational study with prospective data collection conducted in maternal patients with premature rupture of membranes (PROMs) for more than 12 h and meconium-stained amniotic fluid (MSAF) and their infants at a 400-bed public referral hospital in Surabaya. Specimens were collected in the delivery and operating rooms, with additional data collection in the maternal and neonatal care units.

A total of 30 mothers and their newborns were included in this study after giving informed consent. An antibiotic sensitivity test for the amniotic fluid began with a gynecologist collecting 20 to 30 mL immediately after the uterus was opened during labor. The gynecologist collected the specimen in a sterile area during the incision. It was immediately sealed, placed in a padded envelope, and sent to the lab. If the microbiology test showed bacterial growth that looked like contamination, the microbiologist ruled it out and considered it sterile. The viability of microorganisms was maintained during transport, and the fluid was cultured for growth. Sensitivity testing was performed using automated systems (VITEK). Data on antibiotic use and other relevant information were extracted from the patients’ medical records. Data were analyzed using Microsoft Excel and SPSS 29.0 software, and the results are presented in tables as descriptive analyses.

## 5. Conclusions

Ampicillin and ampicillin–sulbactam are associated with good outcomes in mothers and newborns. Gynecologists are encouraged to prescribe the narrow-spectrum antibiotic ampicillin over the broad-spectrum ampicillin–sulbactam. The use of broad-spectrum antibiotics shows a yearly increasing trend. An antibiotic prescribing algorithm is needed to control antibiotic use and ensure effective antibiotic stewardship.

## Figures and Tables

**Figure 1 pharmaceuticals-18-00037-f001:**
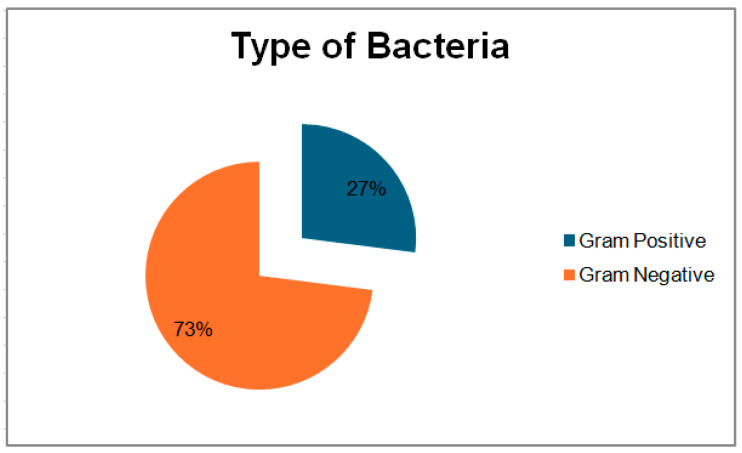
Types of bacteria in positive isolates.

**Figure 2 pharmaceuticals-18-00037-f002:**
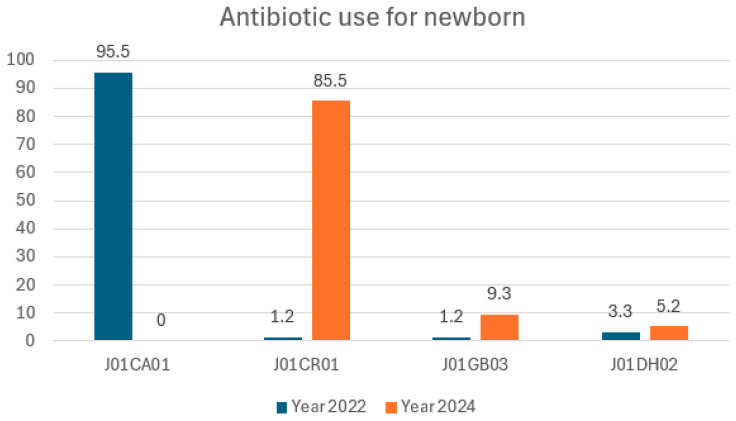
Defined daily dose per 100 bed-days antibiotics in neonates.

**Table 1 pharmaceuticals-18-00037-t001:** Demographic characteristics.

Characteristics (*N* = 90)		N (%)
Maternal Characteristics (*N* = 30)		
Age (Years)	17–25	3 (10)
	26–35	21 (70)
	>35	6 (20)
Gestational Age	Preterm	3 (10)
	Early term	9 (30)
	Full term	13 (43)
	Late term	5 (17)
Parity	Nulliparous	14 (47)
	Multipara	16 (53)
Fetal Malpresentation	Yes	3 (10)
	No	27 (90)
Amniotic Condition	PROMs ^1^ > 12 h	17 (57)
	MSAF ^2^	11 (37)
	PROMs with MSAF	2 (6)
Neonates Characteristics (*N* = 60)		
Gender	Male	37 (62)
Weight	Low	5 (8)
	Normal	55 (92)
Amniotic Condition	MSAF ^2^	41 (68)
CRP ^3^ (mg/L)	>5	7 (12)
Leukocyte (10^3^/μL)	9.00–34.00	60 (100)

^1^ premature rupture of membranes; ^2^ meconium-stained amniotic fluid; ^3^ C-reactive protein.

**Table 2 pharmaceuticals-18-00037-t002:** Laboratory findings and microorganism profiles in maternal samples.

Microorganism	Leukocytes (10^3^/μL)			Neutrophils (%)		
	PROMs ^1^	MSAF ^2^	PROMs ^1^–MSAF ^2^	PROMs ^1^	MSAF ^2^	PROMs ^1^–MSAF ^2^
Gram-positive						
*Staphylococcus haemolyticus*	14.76	-	-	83.40	-	-
*Streptococcus beta haemolyticus*	17.88	-	-	91.40	-	-
*fStaphylococcus epidermis*	-	20.57	-	-	90.70	-
Gram-negative						
*Enterococcus* spp.	15.60	-	-	90.50	-	-
*Enterobacter* spp.	-	15.60	-	-	90.50	-
*Escherichia coli*	15.19	12.24	-	93.80	89.60	-
	-	17.58	-	-	92.02	-
	-	17.90	-	-	91.20	-
*Escherichia coli* and *Klebsiella pneumoniae*	-	10.67	-	-	80.50	-
	-	14.02	-	-	88.60	-
*Escherichia coli ESBL*	-	-	15.52	-	-	85.50
*Escherichia coli ESBL*	-	-	11.11	-	-	72.90

^1^ premature rupture of membranes; ^2^ meconium-stained amniotic fluid.

**Table 3 pharmaceuticals-18-00037-t003:** Laboratory findings and microorganism profiles in neonates.

Microorganism	CRP ^1^ (mg/L)			Leukocytes (10^3^/μL)			Neutrophils (%)		
	PROMs ^2^	MSAF ^3^	PROMs ^2^–MSAF ^3^	PROMs ^2^	MSAF ^3^	PROMs ^2^–MSAF ^3^	PROMs ^2^	MSAF ^3^	PROMs ^2^–MSAF ^3^
Gram-positive									
*Staphylococcus haemolyticus*	-	-	-	-	-	-	-	-	-
*Streptococcus beta haemolyticus*	<5	-	-	22.06	-	-	73.10	-	-
*Staphylococcus epidermis*	-	<5	-	-	33.23	-	-	75.40	-
Gram-negative									
*Enterococcus* spp.	<5	-	-	18.31	-	-	75.20	-	-
*Enterobacter* spp.	-	16.00	-	-	8.52	-	-	76.90	-
*Escherichia coli*	<5	-	-	18.93	14.58	-	79.60	67.20	-
	-	15.00	-	-	24.22	-	-	76.90	-
	-	-	-	-	22.49	-	-	79.70	-
*Escherichia coli* and *Klebsiella pneumoniae*	-	13.00	-	-	19.85	-	-	85.10	-
	-	12.50	-	-	20.58	-	-	70.30	-
*Escherichia coli ESBL*	-	-	25.00	-	-	14.87	-	-	85.60
*Escherichia coli ESBL*	-	-	7.50	-	-	-	-	-	-

^1^ C-reactive protein; ^2^ premature rupture of membranes; ^3^ meconium-stained amniotic fluid.

**Table 4 pharmaceuticals-18-00037-t004:** Clinical outcomes based on amniotic fluid conditions in newborns.

Clinical Outcomes	PROMs ^2^ (*N* = 17)	MSAF ^3^ (*N* = 11)	PROMs ^2^–MSAF ^3^ (*N* = 2)
Length of Stay (days) *p* = 0.03			
1–3	10	3	0
>3	6	8	2
APGAR ^1^ Score *p* = 0.687			
First minute	5.4 ± 2.32	5.6 ± 2.33	4 ± 4.24
Fifth minute	6.4 ± 2.32	6.6 ± 2.33	5 ± 4.24
Asphyxia	6	4	0
Meconium Aspiration Syndrome	0	0	1
Weight (kg)	3.1 ± 0.40	3.2 ± 0.39	3.4 ± 0.14
NICU admission	0	0	0

^1^ APGAR: Appearance, Pulse, Grimace, Activity, and Respiration; ^2^ premature rupture of membranes; ^3^ meconium-stained amniotic fluid.

**Table 5 pharmaceuticals-18-00037-t005:** Defined daily dose per 100 bed-days antibiotics in neonates.

AWaRe ^1^	Name	ATC ^2^	Year 2022	Year 2024
Access	Ampicillin	J01CA01	4.19 (95.5)	
Access	Ampicillin/sulbactam	J01CR01	0.05 (1.2)	3.14 (85.5)
Access	Gentamicin	J01GB03	0.05 (1.2)	0.34 (9.3)
Watch	Meropenem	J01DH02	0.14 (3.3)	0.19 (5.2)
	Total		4.38 (100)	3.67 (100)

^1^ The WHO AWaRe classification. ^2^ Anatomical Therapeutic Chemical (a drug classification system used to categorize medications).

## Data Availability

Data are available on request.

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
