# Peer review of "Microbial Pattern in Amniotic Fluid from Women with Premature Rupture of Membranes and Meconium-Stained Fluid"

_pharmaceuticals, 2024, doi:10.3390/ph18010037_

Round 1
Reviewer 1 Report
Comments and Suggestions for Authors
The paper entitled: “Microbial Pattern of Amniotic Fluid in Women with Premature Rupture of Membranes and Meconium-Stained Fluid” by Fauna Herawati et al. describes a clinical observational study performed on a 30 cases of pregnancy in which premature rupture of membranes occurred and amniotic fluid was then analysed for microbiological retrievements.
As such the study is small indeed but could be data could be significant due to their real life meaning and regarding a possibly serious clinical condition.
However, the paper has been submitted to the journal Pharmaceuticals and therefore, pharmaceutically pertinent deductions should be at least proposed or discussed.
Among microbiological retrieves, there are various known pathogens. However, it is unclear how and if these findings can be correlated to eventual or suggested antimicrobial therapies.
Methods are not described in detail. Microbiological methods should at least be provided.
Charts or diagrams about analytical results could be provided in order to improve the graphical impact of the paper.
Statistical analysis is at a minimum and more details should be provided.
Author Response
Comments 1:
However, the paper has been submitted to the journal Pharmaceuticals and therefore, pharmaceutically pertinent deductions should be at least proposed or discussed.
Responses 1: Thank you. I appreciate your consideration.
The author added this information in Introduction section.
Meconium-containing amniotic fluid occurs in approximately 1 in 10 births and is the fetal fecal matter {7-9]. It is crucial to recognize that amniotic fluid can contain ammonium when the fetus experiences stress [10]. Postnatal management must focus on promptly clearing meconium from the baby's airway, as low Apgar scores are directly linked to a significantly higher risk of mortality [7,10-11].
Meconium itself is not a risk factor for infection [7,18-19]; antibiotics should only be administered when there is clear evidence of infection [20]. The use of antibiotics in newborns can lead to serious complications, including bronchopulmonary dysplasia [22] and dysbiosis [23,24], which disrupts immune function and increases the risk of metabolic disorders later in life. This study clearly demonstrates a concerning trend of high broad-spectrum antibiotic use in both mothers and newborns. It is imperative that clinical practice be guided by sound evidence, utilizing antibiotics only for confirmed bacterial infections, rather than across the board in cesarean deliveries with meconium-stained amniotic fluid.
Reference:
[7] Dahiana M. Gallo, Roberto Romero, Mariachiara Bosco, Francesca Gotsch, Sunil Jaiman, Eunjung Jung, Manaphat Suksai, Carlos López Ramón y Cajal, Bo Hyun Yoon, Tinnakorn Chaiworapongsa. Meconium-stained amniotic fluid. American Journal of Obstetrics & Gynecology. 2023; 228 (5): S1158 - S1178
[8] Saint-Fleur AL, Alcalá HE, Sridhar S. Outcomes of neonates born through meconium-stained amniotic fluid pre and post 2015 NRP guideline implementation. PLoS One. 2023;18(8):e0289945. Published 2023 Aug 10. doi:10.1371/journal.pone.0289945
[9] Tolu LB, Birara M, Teshome T, Feyissa GT. Perinatal outcome of meconium stained amniotic fluid among labouring mothers at teaching referral hospital in urban Ethiopia. PLoS One. 2020;15(11):e0242025. Published 2020 Nov 13. doi:10.1371/journal.pone.0242025
[10] Rawat M, Nangia S, Chandrasekharan P, Lakshminrusimha S. Approach to Infants Born Through Meconium Stained Amniotic Fluid: Evolution Based on Evidence?. Am J Perinatol. 2018;35(9):815-822. doi:10.1055/s-0037-1620269
[11] Committee Opinion No 689: Delivery of a Newborn With Meconium-Stained Amniotic Fluid. Obstet Gynecol. 2017;129(3):e33-e34. doi:10.1097/AOG.0000000000001950
[18] Parween S, Prasad D, Poonam P, Ahmar R, Sinha A, Ranjana R. Impact of Meconium-Stained Amniotic Fluid on Neonatal Outcome in a Tertiary Hospital. Cureus. 2022;14(4):e24464. Published 2022 Apr 25. doi:10.7759/cureus.24464
[19] Kelly LE, Shivananda S, Murthy P, Srinivasjois R, Shah PS. Antibiotics for neonates born through meconium-stained amniotic fluid. Cochrane Database of Systematic Reviews 2017, Issue 6. Art. No.: CD006183. DOI: 10.1002/14651858.CD006183.pub2
[20] WHO Recommendations for Prevention and Treatment of Maternal Peripartum Infections: Highlights and Key Messages from the World Health Organization’s 2015 Global Recommendations. Available from: https://iris.who.int/bitstream/handle/10665/186684/WHO_RHR_15.19_eng.pdf?sequence=1.
[21] Manoj Kumar Ram, Bankey Bihari Singh. APGAR scores and perinatal outcome in neonates born with meconium stained amniotic fluid. International Journal of Medical and Health Research. 2021; 7 (2): 15-18
[22] Yu W, Zhang L, Li S, et al. Early Antibiotic Use and Neonatal Outcomes Among Preterm Infants Without Infections. Pediatrics. 2023;151(5):e2022059427
[23] Lamont RF, Møller Luef B, Stener Jørgensen J. Childhood inflammatory and metabolic disease following exposure to antibiotics in pregnancy, antenatally, intrapartum and neonatally. F1000Res. 2020;9:F1000 Faculty Rev-144. Published 2020 Feb 25. doi:10.12688/f1000research.19954.1
[24] Shekhar S, Petersen FC. The Dark Side of Antibiotics: Adverse Effects on the Infant Immune Defense Against Infection. Front Pediatr. 2020;8:544460. Published 2020 Oct 15. doi:10.3389/fped.2020.544460
Comments 2:
Among microbiological retrieves, there are various known pathogens. However, it is unclear how and if these findings can be correlated to eventual or suggested antimicrobial therapies.
Responses 2: The authors added information in Discussion section.
Antibiotics are used against bacteria, therefore aside from indication; antibiotic choices are associated with the type of bacteria and its sensitivity to antibiotics.
Comments 3:
Methods are not described in detail. Microbiological methods should at least be provided.
Responses 3:
The authors replace information in Methods section.
Amniotic fluid samples were collected by obstetricians immediately after the uterus was opened during labor. The samples were then sent to the microbiology laboratory for culture and antimicrobial susceptibility testing.
Revise sentences
An antibiotic sensitivity test for amniotic fluid begins with collecting 20 to 30 ml by obstetricians immediately after the uterus was opened during labor. The viability of microorganisms is maintained during transport and the fluid is cultured for growth, furthermore sensitivity testing is performed using automated systems (VITEK).
Comments 4:
Charts or diagrams about analytical results could be provided in order to improve the graphical impact of the paper.
Responses 4:
The authors added charts below Table 5.
Comments 5:
Statistical analysis is at a minimum and more details should be provided.
Responses 5:
Thank you for the suggestion but this study is intended to identify trends (surveillance) through descriptive statistics, rather than employing the analytical statistics required in a factor analysis.
Reviewer 2 Report
Comments and Suggestions for Authors
The article entitled “Microbial Pattern of Amniotic Fluid in Women with Premature Rupture of Membranes and Meconium-Stained Fluid” is based on investigation of the microbial patterns in the amniotic fluid of women with PROM and MSAF to determine the presence and types of bacterial growth. In addition, this study also emphasizes on the microbial susceptibility towards certain types of antibiotics.
The overall organization and quality of the manuscript is good. However, I have few minor reservation and suggestions that the authors need to correct before the manuscript is being processed onward.
Comments are:
1. A lot of the previously published articles are available relevant to the research reported in this manuscript (in search of identification of microbial stains diagnosed with PROM and MSAF like IAI infections) which limit the worth of this work. Authors are encouraged to justify the novelty and uniqueness of this work.
2. At Page 2, Line 55-56, the sentence mentioned is showing over lapping. Please correct it.
3. The Introduction section furnish limited information about PROM and MSAF like IAI infections. The authors are encouraged to provide brief summary of all the factors that contribute in PROM and MSAF like IAI infections.
4. The descriptions provided in the Introduction section, about bacterial profiles in amniotic fluid during the onset of PROM and MSAF are not sufficient enough. Authors are encouraged to provide some more relevant data about bacterial profiles from the previously published articles.
5. In Material and Method section, authors are encouraged to provide the exact methodology they have adopted to confirm the susceptibility of bacterial isolates to certain antibiotics.
6. In Table-1, while discussing demographic characteristics, please justify the term “N(90)” what (90) is referring to.
7. In Table-5, please provide the full form of ATC in caption is it is confusing.
8. Page 5, Line 146. Please provide the full form of term “OR”.
9. Justify the ROC analysis.
10. The conclusion section is poorly written and lacking the summary of the overall findings. Evaluation regarding particular bacteria involved in IAI infections (including PROM and MSAF) has not been properly explained and future perspectives have not been discussed clearly. Please rewrite the conclusion section.
11. Most of the references are out of format of this journal. For instance, 13, 15, 20, 23, 25 and so on. The authors are encouraged to revisit all the references for the possible corrections such as naming style of the authors, lower hand style of titles, upper hand style of journal names, style of page numbers and so on.
I recommend this manuscript for publication in this journal after incorporating the above mentioned minor corrections.
Author Response
Comment 1:
1. A lot of the previously published articles are available relevant to the research reported in this manuscript (in search of identification of microbial stains diagnosed with PROM and MSAF like IAI infections) which limit the worth of this work. Authors are encouraged to justify the novelty and uniqueness of this work.
Response 1:
Thank you. I appreciate your consideration.
I added this information in Introduction section.
Meconium-containing amniotic fluid occurs in approximately 1 in 10 births and is the fetal fecal matter {7-9]. It is crucial to recognize that amniotic fluid can contain ammonium when the fetus experiences stress [10]. Postnatal management must focus on promptly clearing meconium from the baby's airway, as low Apgar scores are directly linked to a significantly higher risk of mortality [7,10-11].
Meconium itself is not a risk factor for infection [7,18-19]; antibiotics should only be administered when there is clear evidence of infection [20]. The use of antibiotics in newborns can lead to serious complications, including bronchopulmonary dysplasia [22] and dysbiosis [23,24], which disrupts immune function and increases the risk of metabolic disorders later in life. This study clearly demonstrates a concerning trend of high broad-spectrum antibiotic use in both mothers and newborns. It is imperative that clinical practice be guided by sound evidence, utilizing antibiotics only for confirmed bacterial infections, rather than across the board in cesarean deliveries with meconium-stained amniotic fluid.
Reference:
[7] Dahiana M. Gallo, Roberto Romero, Mariachiara Bosco, Francesca Gotsch, Sunil Jaiman, Eunjung Jung, Manaphat Suksai, Carlos López Ramón y Cajal, Bo Hyun Yoon, Tinnakorn Chaiworapongsa. Meconium-stained amniotic fluid. American Journal of Obstetrics & Gynecology. 2023; 228 (5): S1158 - S1178
[8] Saint-Fleur AL, Alcalá HE, Sridhar S. Outcomes of neonates born through meconium-stained amniotic fluid pre and post 2015 NRP guideline implementation. PLoS One. 2023;18(8):e0289945. Published 2023 Aug 10. doi:10.1371/journal.pone.0289945
[9] Tolu LB, Birara M, Teshome T, Feyissa GT. Perinatal outcome of meconium stained amniotic fluid among labouring mothers at teaching referral hospital in urban Ethiopia. PLoS One. 2020;15(11):e0242025. Published 2020 Nov 13. doi:10.1371/journal.pone.0242025
[10] Rawat M, Nangia S, Chandrasekharan P, Lakshminrusimha S. Approach to Infants Born Through Meconium Stained Amniotic Fluid: Evolution Based on Evidence?. Am J Perinatol. 2018;35(9):815-822. doi:10.1055/s-0037-1620269
[11] Committee Opinion No 689: Delivery of a Newborn With Meconium-Stained Amniotic Fluid. Obstet Gynecol. 2017;129(3):e33-e34. doi:10.1097/AOG.0000000000001950
[18] Parween S, Prasad D, Poonam P, Ahmar R, Sinha A, Ranjana R. Impact of Meconium-Stained Amniotic Fluid on Neonatal Outcome in a Tertiary Hospital. Cureus. 2022;14(4):e24464. Published 2022 Apr 25. doi:10.7759/cureus.24464
[19] Kelly LE, Shivananda S, Murthy P, Srinivasjois R, Shah PS. Antibiotics for neonates born through meconium-stained amniotic fluid. Cochrane Database of Systematic Reviews 2017, Issue 6. Art. No.: CD006183. DOI: 10.1002/14651858.CD006183.pub2
[20] WHO Recommendations for Prevention and Treatment of Maternal Peripartum Infections: Highlights and Key Messages from the World Health Organization’s 2015 Global Recommendations. Available from: https://iris.who.int/bitstream/handle/10665/186684/WHO_RHR_15.19_eng.pdf?sequence=1.
[21] Manoj Kumar Ram, Bankey Bihari Singh. APGAR scores and perinatal outcome in neonates born with meconium stained amniotic fluid. International Journal of Medical and Health Research. 2021; 7 (2): 15-18
[22] Yu W, Zhang L, Li S, et al. Early Antibiotic Use and Neonatal Outcomes Among Preterm Infants Without Infections. Pediatrics. 2023;151(5):e2022059427
[23] Lamont RF, Møller Luef B, Stener Jørgensen J. Childhood inflammatory and metabolic disease following exposure to antibiotics in pregnancy, antenatally, intrapartum and neonatally. F1000Res. 2020;9:F1000 Faculty Rev-144. Published 2020 Feb 25. doi:10.12688/f1000research.19954.1
[24] Shekhar S, Petersen FC. The Dark Side of Antibiotics: Adverse Effects on the Infant Immune Defense Against Infection. Front Pediatr. 2020;8:544460. Published 2020 Oct 15. doi:10.3389/fped.2020.544460
Comment 2:
2. At Page 2, Line 55-56, the sentence mentioned is showing over lapping. Please correct it.
Response 2:
The authors delete the over lapping sentence: ‘Overall, 70% (7/10) of the prominent.’
Comment 3:
3. The Introduction section furnish limited information about PROM and MSAF like IAI infections. The authors are encouraged to provide brief summary of all the factors that contribute in PROM and MSAF like IAI infections.
Response 3:
The authors added information in the Introduction section. (see Response 1)
Comment 4:
4. The descriptions provided in the Introduction section, about bacterial profiles in amniotic fluid during the onset of PROM and MSAF are not sufficient enough. Authors are encouraged to provide some more relevant data about bacterial profiles from the previously published articles.
Response 4:
The authors added information in the Introduction section. (see Response 1)
Comment 5:
5. In Material and Method section, authors are encouraged to provide the exact methodology they have adopted to confirm the susceptibility of bacterial isolates to certain antibiotics.
Response 5:
The authors replace information in Methods section.
Amniotic fluid samples were collected by obstetricians immediately after the uterus was opened during labor. The samples were then sent to the microbiology laboratory for culture and antimicrobial susceptibility testing.
Revise sentences
An antibiotic sensitivity test for amniotic fluid begins with collecting 20 to 30 ml by obstetricians immediately after the uterus was opened during labor. The viability of microorganisms is maintained during transport and the fluid is cultured for growth, furthermore sensitivity testing is performed using automated systems (VITEK).
Comment 6:
6. In Table-1, while discussing demographic characteristics, please justify the term “N(90)” what (90) is referring to.
Response 6: Thank you for your consideration. The author writes (N=90) refer to 90 mothers.
Comment 7:
7. In Table-5, please provide the full form of ATC in caption is it is confusing.
Response 7: Thank you for your consideration.
The author added in the footnote of the Table 5: Anatomical Therapeutic Chemical (a drug classification system used to categorize medications)
Comment 8:
8. Page 5, Line 146. Please provide the full form of term “OR”.
Response 8: Thank you for your consideration. The author added information ‘(Odd Ratio)’ in that sentence.
Comment 9:
9. Justify the ROC analysis.
Response 9: Thank you for your consideration. The author revises the sentence.
Previous
ROC analysis showed a significant increase in the risk of neonatal sepsis after 37 hours of latency
Revise sentence
Neonatal sepsis rates increase after 37 hours of (PROM) latency.
Comment 10:
10. The conclusion section is poorly written and lacking the summary of the overall findings. Evaluation regarding particular bacteria involved in IAI infections (including PROM and MSAF) has not been properly explained and future perspectives have not been discussed clearly. Please rewrite the conclusion section.
Response 10: Thank you for your consideration. The author revises the sentence.
Previous
The highest incidence of bacterial growth was seen in patients with meconi-um-stained amniotic fluid (MSAF), where 7 out of 11 samples were positive for bacteria. The antibiotics with the highest sensitivity (11/11, 100%) among all isolates were amoxicillin-clavulanic acid, piperacillin-tazobactam, amikacin, chloramphenicol, and meropenem.
Revise sentence
Ampicillin and ampicillin-sulbactam are associated with good outcomes in mothers and newborns. Gynecologists are encouraged to prescribe the narrow-spectrum antibiotic ampicillin over the broad-spectrum ampicillin-sulbactam.
Comment 11:
11. Most of the references are out of format of this journal. For instance, 13, 15, 20, 23, 25 and so on. The authors are encouraged to revisit all the references for the possible corrections such as naming style of the authors, lower hand style of titles, upper hand style of journal names, style of page numbers and so on.
Response 11: Thank you for your consideration. The author revises the references format.
Round 2
Reviewer 1 Report
Comments and Suggestions for Authors
From our previous revision, unanswered issues still remain;
1)Among microbiological retrieves, there are various known pathogens. However, it is unclear how and if these findings can be correlated to eventual or suggested antimicrobial therapies.
>>They added pharmacological deductions about the use of wide-spectrum antibiotics. It is unclear to us why then performing this study. Please specify in the introduction.
2)Methods are not described in detail. Microbiological methods should at least be provided.
>>they improved details of the sampling method. We also like to know more about microbiological testing because this can impact the type of microbial findings. At least discuss possible contaminations during sampling and if microbial resistance testing was performed.
The use of wide-spectrum antibiotics doesn't require specific microbial testing but so why this has been performed here. Please discuss..
3)Charts or diagrams about analytical results could be provided in order to improve the graphical impact of the paper.
>>They added a diagram but some output from the lab could help
4)Statistical analysis is at a minimum and more details should be provided.
>>No statistic used in practice. This can be accepted if no correlation is searched and cases are dealt with as independent ones. But this must be stated in the abstract and introduction.
Author Response
Reviewer comment:
1)Among microbiological retrieves, there are various known pathogens. However, it is unclear how and if these findings can be correlated to eventual or suggested antimicrobial therapies.
>>They added pharmacological deductions about the use of wide-spectrum antibiotics. It is unclear to us why then performing this study. Please specify in the introduction
Author response:
Thank you for your valuable feedback to enhance the clarity of our manuscript. We have clarified the purpose of our study in the introduction and included the following information:
Previous paragraph
The presence of meconium in the amniotic fluid may facilitate bacterial growth by acting as a growth medium, potentially inhibiting the natural bacteriostatic properties of the fluid or compromising immune defense mechanisms, thereby increasing the risk of IAI [12]. The most accepted route of intra-amniotic infection is the ascent of microorganisms from the lower genital tract [4]. Most women with intra-amniotic infection had common vaginal bacteria in their amniotic fluid, dominated by Ureaplasma urealyticum, Lactobacillus, Escherichia coli, and Streptococcus agalactiae [13]. A cross-sectional study of women with intra-amniotic infection and intact membranes found that 75% (6/8) had bacteria in their amniotic fluid typical of the vaginal ecosystem. Of these, 62.5% (5/8) also had the same bacteria in their vagina, including Ureaplasma urealyticum, Escherichia coli, and Streptococcus agalactiae. In addition, 16S rRNA gene sequencing revealed bacterial profiles in the amniotic fluid dominated by species commonly found in the vagina, such as Sneathia, Ureaplasma, and Prevotella. Overall, 70% (7/10) of the prominent operational taxonomic units in amniotic fluid matched those in the vagina, suggesting that ascending bacteria from the lower genital tract is the primary pathway for intra-amniotic infection [13]. A study by Abate et al. showed that pregnant women with meconium-stained amniotic fluid (MSAF) had a significantly higher prevalence of positive amniotic fluid cultures obtained by amniocentesis than women with clear amniotic fluid. Neonates born with MSAF had a 2.5 times higher risk of mortality compared to those born with clear amniotic fluid. The incidence of neonatal sepsis was also higher in neonates born through MSAF (5.6%) compared to those born through clear or colorless amniotic fluid (1%) [14]. Pregnant women with MSAF and PROM are at increased risk of infection, necessitating the administration of antibiotic therapy to both mothers and newborns during delivery [15][16].
Revise paragraph
The presence of meconium in the amniotic fluid may facilitate bacterial growth by acting as a growth medium, potentially inhibiting the natural bacteriostatic properties of the fluid or compromising immune defense mechanisms, thereby increasing the risk of IAI [12]. This infection typically occurs due to the ascent of microorganisms from the lower genital tract [4]. Studies show that common vaginal bacteria, such as Ureaplasma urealyticum, Escherichia coli, and Streptococcus agalactiae, are often found in the amniotic fluid of women with IAI [13]. Pregnant women with meconium-stained amniotic fluid (MSAF) have a higher prevalence of positive amniotic fluid cultures and an increased risk of neonatal sepsis and mortality [14]. Therefore, antibiotic therapy is recommended for both mothers and newborns during delivery in cases of MSAF and premature rupture of membranes (PROM) [15,16].
Previous paragraph
However, the judicious use of antibiotics in newborns is crucial, as excessive or in-appropriate use can lead to antibiotic resistance. As stated in Charles Darwin's theory of evolution, the natural selection occurs when organisms with traits that are better suited to their environment are more likely to survive and reproduce, passing these beneficial traits on to the next generation. This natural selection is known as selective pressure in the development of antibiotic resistance in bacteria, driving the evolution of species over time [17]. Antibiotics combat bacteria, so identifying the specific bacteria causing an infection is essential. This understanding guides appropriate antibiotic use, prevents overuse of wide-spectrum antibiotics, and minimizes the risk of antibiotic resistance. Meconium itself is not a risk factor for infection [1,18-20]; antibiotics should only be administered when there is clear evidence of infection [21]. The use of antibiotics in newborns can lead to serious complications, including bronchopulmonary dysplasia [22] and dysbiosis [23,24], which disrupts immune function and increases the risk of metabolic disorders later in life. This study clearly demonstrates a concerning trend of high broad-spectrum antibiotic use in both mothers and newborns. It is imperative that clinical practice be guided by sound evidence, utilizing antibiotics only for confirmed bacterial infections, rather than across the board in cesarean deliveries with meconium-stained amniotic fluid.
Revise paragraph:
Judicious use of antibiotics in newborns is imperative to prevent antibiotic resistance. According to Darwin's theory of evolution, natural selection favors traits that enhance survival, leading to antibiotic resistance in bacteria [17]. Identifying the specific bacteria causing an infection guides appropriate antibiotic use, preventing overuse of broad-spectrum antibiotics and minimizing resistance risk. Meconium itself is not a risk factor for infection [1,18-20]; antibiotics should only be administered when there is clear evidence of infection [21]. A review reported that narrow or broad spectrum antibiotics have similar efficacy in endometritis, febrile morbidity, wound infection, and urinary tract infection in mother after postoperative cesarean sections. However, there is no data available on infant outcomes [22]. Overuse in newborns can cause serious complications like bronchopulmonary dysplasia [22] and dysbiosis [23,24], which disrupt immune function and increase the risk of metabolic disorders. This study highlights the concerning trend of high broad-spectrum antibiotic use in mothers and newborns, emphasizing the need for evidence-based clinical practice to use antibiotics only for confirmed infections.
Reviewer comment:
2)Methods are not described in detail. Microbiological methods should at least be provided.
>>they improved details of the sampling method. We also like to know more about microbiological testing because this can impact the type of microbial findings. At least discuss possible contaminations during sampling and if microbial resistance testing was performed.
Author response:
Thank you for your suggestion to improve the manuscript.
We added information in Methods section:
The gynecologist collected the specimen in a sterile area during the incision. It was immediately sealed, placed in a padded envelope, and sent to the lab. If the microbiology test shows bacterial growth that looks like contamination, the microbiologist rules it out and considers it sterile.
Reviewer comment:
The use of wide-spectrum antibiotics doesn't require specific microbial testing but so why this has been performed here. Please discuss.
Author response:
Thank you for your suggestion to improve the manuscript.
As reported in the Results section, only 43% of isolates grew, which is a limitation of this study. These microbiological examinations were performed because prescribers preferred broad-spectrum antibiotics to narrow-spectrum antibiotics. These microbio-logical findings will encourage prescribers to use narrow-spectrum antibiotics.
We added information in Discussion section:
As reported in the Results section, only 43% of isolates grew, which is a limitation of this study. This small number of positive isolates in microbiological findings will encourage prescribers to use narrow-spectrum antibiotics and develop clinical prediction for antibiotic use.
Reviewer comment:
3)Charts or diagrams about analytical results could be provided in order to improve the graphical impact of the paper.
>>They added a diagram but some output from the lab could help
Author response:
Thank you for your suggestion to improve the manuscript.
We added ‘Figure 1. Types of bacteria in positive isolates.’ in the Results section.
Reviewer comment:
4)Statistical analysis is at a minimum and more details should be provided.
>>No statistic used in practice. This can be accepted if no correlation is searched and cases are dealt with as independent ones. But this must be stated in the abstract and introduction.
Author response:
Thank you for your suggestion to improve the manuscript.
We added this information in the abstract and at the end of the Introduction section.
Abstract
This study identifies the microbial patterns in the amniotic fluid of women with PROM and MSAF to determine the presence and types of bacterial growth also identifies trends in antibiotic use through descriptive statistics.;
Introduction
This surveillance study identifies trends in antibiotic use through descriptive statistics. A correlation analysis was not performed as the study was not designed to identify infection risk factors.